# "It could bring a lot of help to people that aren't getting help right now": A qualitative analysis of the impact of virtual care on access to primary care for people with opioid use disorder

Shawna Narayan[1], Sarah Spencer [2], Lindsay Hedden[2]*, Ellie Gooderham[2], Sarah Muñoz-Violant [2], Rita K. McCracken [1,3,4]*

1 Faculty of Medicine, University of British Columbia, Vancouver, British Columbia, Canada, 2 Faculty of Health Sciences, Simon Fraser University, Burnaby, British Columbia, Canada, 3 Department of Family Medicine, Providence Health Care, Vancouver, British Columbia, Canada, 4 School of Medicine, Simon Fraser University, Surrey, British Columbia, Canada

* lindsay_hedden@sfu.ca (LH); rita.mccracken@ubc.ca (RKM)

## Abstract

Primary care plays a vital role for people with opioid use disorder and the COVID-19 pandemic introduced new challenges for this population to access primary care. While the expansion of virtual care was intended to support access to essential primary care, little is known about how this has affected the ability of people with opioid use disorder to access the full range of primary care services. Our objective is to explore the experiences of family physicians and people with opioid use disorder to understand how virtual care has impacted primary care access for people with opioid use disorder in British Columbia, Canada. We conducted semi-structured interviews with people with opioid use disorder and licensed family physicians who provide care to this population. Interviews were recorded, transcribed, and thematically analyzed. We present participants' experiences by the six dimensions of access: availability, accessibility, accommodation, affordability, acceptability, and awareness. Virtual visits were perceived by participants as a valuable option for accessing primary care needs, in addition to opioid use disorder management. These data describe how tailoring virtual care to the complex and intersecting needs of people with opioid use disorder can help promote equitable access to primary care. Our findings highlight the varied dimensions of access that influence virtual care experiences for people with opioid use disorder and their primary care providers. As virtual care becomes a routine primary care modality, ensuring equitable access will require attention to each dimension.

**Data availability statement:** In accordance with the PLOS data access policy, our ethical obligation to research that upholds participant confidentiality, and the inability to fully anonymize the data collected and analyzed for this research, de-identified data may be made available on request to researchers, subject to approval from the REB of record (Simon Fraser University, dore@sfu.ca).

**Funding:** This work was supported by the Canadian Institutes for Health Research (WI1-179888 to LH and RKM) and Michael Smith Health Research BC (to LH). The funder had no role in study design, data collection and analysis, decision to publish, or preparation of the manuscript.

**Competing interests:** The authors have declared that no competing interests exist.

## Author summary

The COVID-19 pandemic drove a rapid and near-universal introduction of virtual visits in primary care in Canada. As a public health measure, virtual care provides a means to minimize exposure risks but, as a modality of delivering primary care, virtual care may compromise access for those who do not have the requisite technology or private spaces to accommodate confidential medical consultations. This is particularly salient amongst people with opioid use disorder who frequently face concurrent challenges of poverty, housing insecurity, and multiple ongoing health concerns. This not only compromises opioid use disorder care but also management of individuals' overall health and wellbeing. Given the persistence of virtual care as a core primary care delivery modality beyond the pandemic, we conducted interviews with family physicians and people with opioid use disorder to understand how virtual care has impacted access to primary care. Our results demonstrate that virtual modalities can be a viable and desirable modality for accessing primary care. To promote more equitable access to primary care, however, family physicians must tailor their use of virtual care to the varied needs of people with opioid use disorder.

## Introduction

When British Columbia (BC), Canada declared COVID-19 a public health emergency in March 2020 [1], the province was already in the fourth year of its toxic opioid-related overdose public health emergency [2]. The co-occurrence of these crises strained healthcare service delivery [3] and exacerbated health inequities [4], especially for people with opioid use disorder (PWOUD) [5]. In response to COVID-19, non-essential healthcare services were reduced, primary care clinics temporarily closed or restricted access, and providers shifted to synchronous virtual modalities to provide care by phone or video [6,7].

Across Canada, access to primary care is an ongoing challenge, worsened by the pandemic. 17.2% of Canadians reported they do not have a regular provider in 2023 (compared to 14.5% in 2019) [8,9]. Further, social location (e.g., income, housing, racialization) frequently affects access to primary care [8,10]. Low-socioeconomic status, rural, racialized, and sex and gender minority communities disproportionly face access barriers such as long wait times, transportation difficulties, and culturally unresponsive care [11–13]. These barriers in access can have serious consequences resulting from delayed care and unmanaged conditions [14–16].

PWOUD face additional inequities with intersecting impacts of social exclusion, discrimination, criminalization, stigma, poverty, and housing insecurity [17]. Despite ongoing increases in the number of family physicians (FPs) per capita [18], equitable access to primary care has deteriorated in Canada, potentially due to reforms that fail to respond to the complexities of marginalized communities [10,19].

For PWOUD, regular access to primary care is essential. In addition to managing opioid agonist therapy (OAT), primary care plays a vital role for PWOUD in managing prevalent comorbid conditions such as respiratory illnesses [20], sexually transmitted and blood-borne infections [21], and mental health disorders [22]. With FPs at the forefront of managing opioid use disorder (OUD) and chronic conditions [23], ensuring access to primary care can improve health outcomes for PWOUD [24,25].

Little is known about how the expansion of virtual care has affected the ability of PWOUD to access comprehensive primary care [26]. The broader body of research indicates that virtual modalities can support improved access to primary care for some patients, but several factors (e.g., the digital divide, health literacy, patient demographics and social location) may impede equity of access to virtual care offerings [27–29]. Research focused on virtual care for PWOUD primarily focuses on OUD management itself [30–33], with limited attention to the comprehensive primary care needs of this population and their experiences with virtual care, and often only engages with health care professionals [33–41]. The perspectives of PWOUD are essential to inform equitable care delivery models, especially as virtual care becomes a regular feature of the health system. Thus, our objective is to explore the virtual care experiences of FPs and PWOUD to understand how virtual care has impacted primary care access for PWOUD in BC.

## Methods

### Study design

This analysis was conducted as part of a larger mixed-method study that seeks to characterize changes to primary care access and patient outcomes following the rapid introduction of virtual care for PWOUD [42]. As part of that study, we conducted qualitative interviews with FPs and PWOUD across BC.

### Sampling and recruitment

Eligible PWOUD participants self-reported having received a diagnosis of opioid-use disorder, treatment for opioid use disorder, or having attended a withdrawal management (detox) program for opioid use. We recruited PWOUD participants using targeted outreach provided by study partners and through dissemination of study posters to community-based services and substance-use organizations. We invited participants to share study information (i.e., recruitment posters, contact information) with others who may be interested. We followed a maximum variation approach to seek representation by gender, age, and community size [43].

Eligible FP participants had an active license to practice in BC or were enrolled in the Addiction Medicine or Family Medicine Residency training programs and had experience providing primary care to PWOUD. We recruited FPs using multiple strategies (e.g., professional organization outreach, social media, snowball sampling) aiming to achieve maximum variation in FP characteristics (e.g., career stage, gender, practice model, community size) [43].

We continued with recruitment until we reached data sufficiency (i.e., enough data to support a rigorous analysis) [44,45]. We assessed sufficiency of FP and PWOUD data independently based on a combination of the amount of data and variation of experiences within the data, including disconfirming and discrepant cases [44,46].

### Data collection and analysis

Research staff (SN, SS, EG, as well as a cultural safety specialist and interview facilitator) conducted semi-structured interviews from March to October 2023 with PWOUD and FPs via telephone, Zoom videoconference (Zoom Video Communications Inc.), or in-person depending on participant preference. We followed a semi-structured interview guide for each participant group (see S1 File, S2 File). PWOUD interviews explored experiences with accessing virtual care, their perceptions of the impact of virtual visits on their primary care experiences, access to OAT, and treatment for any co-occurring conditions. FP interviews explored perceptions of the accessibility of virtual care approaches (video v. phone visits) for PWOUD, the virtual management of OUD and related co-morbidities, and impact of virtual care on workflow, capacity, and work satisfaction. Interviews were audio recorded, transcribed verbatim, and verified by interviewers.

We analyzed data thematically, guided by a pragmatic approach [47,48]. First, researchers (SN, SS, EG, SMV, and LH) independently reviewed all transcripts and interview notes to familiarize themselves with the data. We then proceeded to independently code the same transcript to develop initial coding frameworks, after which we met to compare codes and coding decisions and to develop a single harmonized codebook. We then piloted the initial codebook with five additional transcripts, alternating between FP and PWOUD interviews, while iteratively updating and refining the codes to ensure consistency of code application. Once the harmonized codebook (see S3 File) was finalized, three researchers (SN, SS, and SMV) coded all transcripts using NVivo V.14 (QSR International). We discussed any coding disagreements until we reached consensus.

For this focused analysis on the impact of virtual care on access to primary care, we use Saurman's theory of access [49] to deductively organize and analyze a subset of data that had previously been inductively coded using our harmonized codebook. In this theory of access, Saurman builds on Penchansky and Thomas's [50] original theory, adding a sixth dimension of 'awareness' [49] to the five dimensions previously conceptualized (accessibility, availability, acceptability, affordability, and adequacy); all six dimensions are defined in Fig 1.

### Ethics and study rigour

We obtained approval for this study from Research Ethics BC (H22-00585). All participants provided either written or verbal informed consent prior to an interview. Consent that was provided verbally was audio recorded and documented by the interviewer, per the behavioural research ethics board-approved study protocol, for phone interviews with PWOUD participants to ensure equitable access for all study participants regardless of their technological capabilities. All data has been de-identified and direct quotes reported throughout this manuscript are attributed using participant codes.

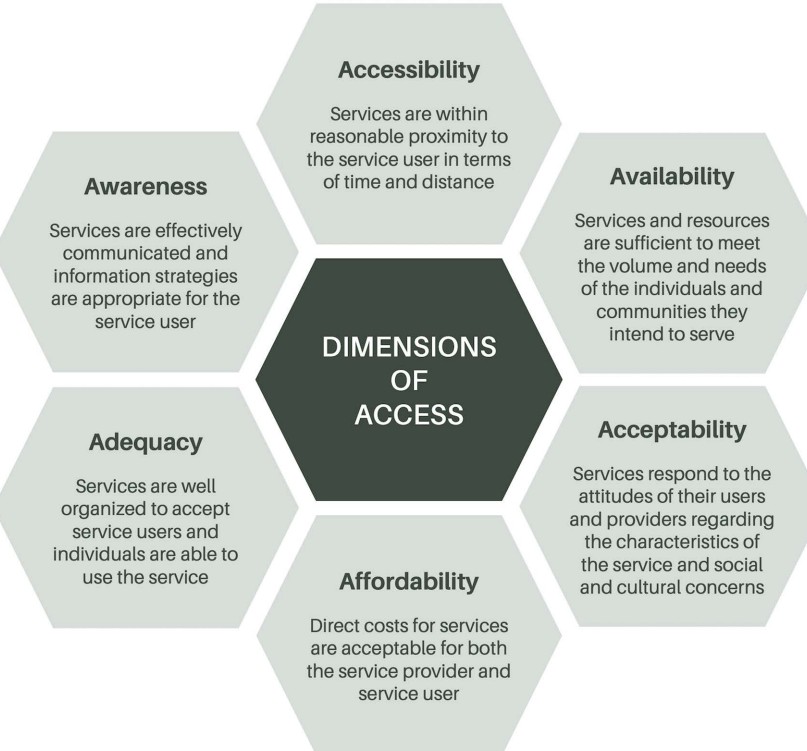

**Fig 1. Dimensions of access [49].**

Our research team included researchers and clinicians with extensive knowledge of primary care and substance use management in BC, allowing us to adopt a pragmatic approach to conduct this research. We prepared interview guides in consultation with health system and substance use researchers, family physicians, and PWOUD. We used experienced interviewers (SN, SS, EG) who took notes, conducted member checking, and confirmed participant meaning during interviews. We maintained detailed records of the iteratively developed codebook and documented discussions, including coding disagreements and their resolution. We searched for negative cases throughout analyses and present results using rich descriptions. Throughout the study, we encouraged self-reflection among all members of the research team [51]. We report our findings using the Standards for Reporting Qualitative Research [52].

## Results

We conducted interviews with 12 physicians and 27 PWOUD (Table 1). Interviews ranged from 10 to 87 minutes in length (Average: FP - 53 minutes; PWOUD - 37 minutes). Most of our participants (9 FPs [75.0%] and 23 PWOUD [85.2%]) practiced and/or lived in an urban location. While all FPs (12 [100.0%]) had at least some experience providing virtual care, just over half (16 [59.3%]) of our PWOUD participants had ever experienced a virtual appointment. We present participants' experiences by the six dimensions of access: accessibility, availability, acceptability, affordability, adequacy, and awareness [49].

### Accessibility

Participants routinely reflected on the convenience of virtual appointments, addressing geographic and logistical barriers to accessing primary care that occur with in-person appointments. This was particularly salient for FP participants who provide care to patients living in more rural settings or who have moved away from the community in which the FP practices and are struggling to find a new primary care provider:

> … patients who move away, virtual care allows them to continually access me from a big distance … just knowing that accessing primary care is a lot more challenging, especially in BC right now, I will hang on to patients for longer. Sometimes I'll say three months… Like I have a couple of patients that, as exceptions, I have continued to keep on my attachment list despite them living on Vancouver Island or somewhere very far away. But these are kind of exceptions that I've made based on the burden of their chronic illnesses, and the fact that I do truly think that they have been looking for primary care and haven't been able to access it. [FP11]

FPs felt that appointment follow-ups, medication reassessments, and mental health concerns could often be addressed effectively by phone, and could be beneficial for PWOUD who encounter transportation barriers:

> Many follow-ups where I've seen somebody in-person, and I suggested they try something for their symptom, and I just want to find out if it worked or not, then I'll talk to them on the phone for a reassessment. So those kinds of things. And yeah, often for patients who are socially anxious or isolated, have difficulty with transportation or mobility, phone visits work well for those. Mental health often works well on the phone. [FP04]

PWOUD participants similarly recognized the value of virtual primary care in its potential to eliminate the commute to their clinic, and that this could afford them greater balance and fewer disruptions to their work, caregiving, or other responsibilities:

> There's so many people who hate going to the doctor's office, or won't go to the doctor's office, or are working and can't take the time off to go to an appointment. I mean the whole option of, on your lunch break, doing a virtual call with

**Table 1. Participant Characteristics [N (%)].**

| | Family Physicians N=12 | People with Opioid Use Disorder N=27 |
|---|---|---|
| **Gender/Sex[a]** | | |
| Woman/female | 10 (83.3) | 8 (29.6) |
| Man/male | 2 (16.7) | 18 (66.7) |
| Undisclosed | 0 (0.0) | 1 (3.7) |
| **Community Size[b]** | | |
| Small urban | 3 (25.0) | 4 (14.8) |
| Urban | 9 (75.0) | 23 (85.2) |
| **Age[c]** | | |
| Mean (SD) | – | 45.2 (11.2) |
| **Years in Primary Care Practice[d]** | | |
| Mean (SD) | 12.1 (10.0) | – |
| **Experience with Virtual Care** | | |
| None | 0 (0.0) | 11 (40.7) |
| Phone only | 2 (16.7) | 11 (40.7) |
| Video only | 0 (0.0) | 1 (3.7) |
| Video and phone | 10 (83.3) | 4 (14.8) |
| **Primary Care Practice Setting[e]** | | |
| Community-based clinic | 12 (100.0) | – |
| Addiction clinic | 4 (33.3) | – |
| Home visits | 1 (8.3) | – |
| Hospital-based practice | 2 (16.7) | – |
| **Primary Care Attachment[f]** | | |
| Attached to a clinic | – | 11 (40.7) |
| Attached to a provider | – | 13 (48.2) |
| Unattached | – | 3 (11.1) |

[a]Gender was asked as an open-ended question; we group gender and sex here to reflect participants' responses using a mix of gender and sex-based language.

[b]Small urban = 10,000 – 99,000; Urban >100,000; FPs reported the community setting in which they provide primary care; PWOUD reported the community setting in which they lived.

[c]Only PWOUD participants were asked their age.

[d]Only FP participants were asked their years in primary care practice.

[e]Only FPs were asked to describe their practice setting; many FP participants provided primary care across multiple settings so these numbers exceed the FP participant counts.

[f]Only PWOUD participants were asked to describe their primary care attachment.

your doctor, taking a 10-minute break to do a virtual call is so much better. And I think so much more people will access healthcare services if they have that option. I think there's a lot of people that are taking advantage of it being virtual; I know I'm one of them. [PWOUD13]

For PWOUD participants living with physical disabilities, getting to (in-person) medical appointments can be challenging, as was detailed by one participant living with Multiple Sclerosis:

In the morning, I can't even get out of bed unless I have opiates in me to start with. So until I get that in me, then I can't walk.... I never knew how much pain was involved with MS until I got it … This morning … I got off a bus, and the bus

driver didn't lower the front … And when I took the step down, I fell over... But because I was a street person, that's what they did. [PWOUD10]

For this participant, the prospect of virtual care was enticing as they felt it would provide a much-desired physical reprieve by eliminating the commute to their clinic (*"I [wouldn't] have to travel"* [PWOUD10]). When asked how virtual care might impact their health and wellbeing, they also felt it would lead to more frequent interactions with their FP (*"I would talk to my doctor a lot more"* [PWOUD10]).

### Availability

Availability of any primary care was a persistent concern amongst FP and PWOUD participants. Notably, virtual care is unable to improve access if broader primary care remains unavailable. PWOUD participants cited long wait times for appointments, limited provider capacity, and lack of attachment to a primary care provider or clinic. Some PWOUD participants attributed these challenges to the COVID-19 pandemic: *"Ever since COVID started, it's impossible to get in … Like, there's lineups"* [PWOUD01]. Others indicated that they lacked a family doctor altogether, despite having care needs beyond managing their OUD: *"Well, I would like a doctor for … my addiction. But also I do have other physical ailments, too, that really need to be addressed"* [PWOUD23]. The lack of primary care attachment often meant PWOUD were reliant on addiction-focused clinics to manage their OUD.

Yet, as FPs who work in addiction-focused clinics shared, their care in these settings is often limited to substance use management. Recognizing the barriers to primary care that their patients face, FP participants indicated they had practiced beyond their clinic's remit (to the extent that resources enable) to ensure patients received the primary care they needed:

So the big problem is that our OAT clinic is just an OAT clinic and doesn't provide any primary care, so those patients would have to be redirected. We do our best, but… Me and the other physician would be able to support some of their other care needs, but very limited. So we don't have any wound care … we can't swab a wound even there. And the nurse practitioner there has a very limited scope. So, technically she shouldn't even be re-prescribing antidepressants, for instance. [FP13]

These experiences suggest that virtual care may not be able to address access to primary care appointments until availability of primary care resources and care (via provider or clinic attachment) is first achieved.

Even for FPs with attached patients and PWOUD with a consistent primary care provider, virtual care can do a lot but still benefits from supplementary in-person visits:

So for patients that are more complicated, especially if they have complex chronic disease or there's mental health or like cognitive issues, then those are the people that I definitely need to see in-person. Like I'm just thinking one, you know, patient who is outreached all the time by nursing, but you can't have these great coherent phone calls with her. And then the nurses are like, 'Oh, she's falling, and her legs are swollen. And this has happened.' So it's when there's these complex cases, those are ones that it's like I need to see the patient in-person. [FP03]

A key aspect of this continued reliance on in-person appointments was the value of FPs being able to use their full range of diagnostic tools, particularly 'laying eyes on' individuals and exploring issues that they may not identify without visual cues. This is likely related to the dominance of phone-based virtual appointments, as most video-based appointments have been abandoned due to inconvenience and the digital divide – both perceived and actual.

## Acceptability

Perceptions of the acceptability of virtual primary care varied depending on the context of its use. Both FP and PWOUD participants highlighted how virtual visits could support care retention. Such uses of virtual care also allow PWOUD to minimize their presence in environments where active substance use is visibly occurring, making virtual appointments more attractive and sometimes beneficial than in-person visits to their primary care provider:

> If they have to go to a physical space that's embedded in an environment where there's active substance use, it can be so triggering and so challenging. So to have the flexibility in those contexts to say, 'You don't need to come see me and step over someone who's using opioids and has nodded off on the street. You don't need to do that. I'll just call you.' [FP08]

Some PWOUD participants felt that virtual appointments could be more comfortable, particularly when discussing stigmatizing experiences or feelings: *"Sometimes I think it's easier … and more comfortable through virtual appointments. Sometimes it's harder to talk about things or bringing things up in-person… Probably about the stigma and stuff"* [PWOUD11]. Yet other PWOUD participants indicated that they needed an in-person appointment to have those difficult and honest conversations with their primary care provider:

> … talking about your mental state, you're going to have a more honest conversation with somebody who is reading your body language and can pick up the vibes that are in the room… the quality of appointment virtually is lower for that amount of human connectivity… [PWOUD19]

Regardless of their experiences with virtual care, PWOUD participants recognized the limits of virtual visits and the need for in-person visits to establish and maintain rapport: *"I like to meet the person first that I'm talking, to establish a rapport, and then it's all fine from there."* [PWOUD23]. This feeling was echoed by FPs who indicated that there were aspects of an in-person visit that could not be replicated over the phone or through a video call:

> … when you're talking to someone on the phone and they say, "Yeah, I'm doing really well, I'm not using, I'm clean," it's impossible to really know, right? I mean we have great rapport, and I do have a lot of trust. But sitting down with someone and looking at them, and just the connection is so much better, too... So I feel like there's a much deeper connection that's still had in-person…I feel like we're missing a lot of that - those cues - on the telephone. [FP09]

PWOUD and FP participants both also emphasized the humanizing aspects of care, highlighting the importance of relational and physical interactions. For example, one participant described how important it was for their physician to see the urgency in their face to appreciate and honour the care that they needed and how this would be lost during interactions via phone:

> …to look in somebody's eyes, it's a lot of communication, and body language, and stuff like that. I feel like… if I hadn't been able to have face-to-face communication with a couple of people that at least did anything for me, they pretty much saved my life… it would have been so easy for them to turn down my request if they didn't see the urgency in my face and see how sick I was. [PWOUD13]

Privacy concerns can impact whether virtual care services were perceived as acceptable. Among PWOUD participants who had some experience with virtual visits, finding a private space to have a confidential conversation with their primary care provider could be a challenge:

> I think having like a phone for that sort of thing in a secure place would be the most ideal. But I'm trying to think of right now where I could even do that. And I can't think of one. [PWOUD04]

Technology and confidentiality also posed a concern for those PWOUD participants who had not experienced and were not interested in video-based virtual care: *"Yeah, it doesn't matter who it is … I'm a little paranoid about security issues, right? Especially virtually, like on the computer, right?"* [PWOUD16]. While FP participants were confident in the security of their virtual connections, they were concerned about their patients having a confidential space to have private medical conversations. This concern was particularly salient for FPs providing care to patients experiencing under-housing or living in communal settings, where they may be using a shared phone to connect with their physician:

> … Because many times a person's handed a phone, and they're waiting for the phone to come back, right? So it's not confidential… I think a lot of times, if it's just someone at the shelter - a staff member who's manning the phones - they're standing there. And we have asked before, 'Is it okay if they go and shut the door, and then they'll come back after?' So there are ways that we've been able to work around it a little bit. But that feels like it's sometimes not happening. [FP09]

**Affordability**

While several of the PWOUD participants had access to a phone, many indicated that the cost of maintaining consistent phone service or internet access could be a challenge. This barrier to virtual care was also recognized by many FP participants: *"The other one is they may not have minutes on their phone, or they may not have a phone. That can be difficult"* [FP03]. PWOUD participants acknowledged that turnover of communication devices could impede physicians' reaching them: *"it's quite often I'm without a phone. And I have tons of voicemail where my doctor's trying to get a hold of me or making appointments and stuff, that I don't know them because I'm unreachable"* [PWOUD11].

Yet, virtual primary care could also pose a more economical option for patients who otherwise would have to travel to see their primary care provider:

> … it makes a lot of sense because people with opioid use disorder have barriers to coming in-person to things, including transportation, including the cost of getting from one place to another, including life being a little bit more chaotic, and schedules being a little bit more unpredictable. [FP04]

Similarly, virtual care can minimize the cost of taking time off work (vis-à-vis lost wages) to attend an in-person appointment. One participant with a blood disorder that required multiple in-person appointments each month expressed their desire for virtual care:

> I haven't had the means to [seek care]. And then I don't want to lose my job. I can't afford to miss one day of work, right?... Because at that point, if I miss one day at work, I'm not eating for two days, right? [PWOUD23]

For FPs, the pandemic enabled important policy shifts such as the introduction of billing codes for virtual visits that further facilitated the rapid adoption of phone and video consultations. Importantly, the new billing codes made it financially viable for family physicians to offer care virtually: *"Having virtual care available for [people who face barriers to care] is a massive improvement for access. And the fact that it's billed equivalently, right?"* [FP04]. However, a few family physicians noted that this transition required an investment in virtual infrastructure, such as secure platforms, additional administrative support, and upgraded phone lines, which added costs for physician owned and operated clinics.

## Adequacy

Among our participants, the ability of virtual care to adequately address the needs of PWOUD varied. Some clinics offered flexible appointment times and a choice of modality (i.e., telephone, video, or in-person), allowing PWOUD to select what worked best for them:

> I'm the driver of that [appointment modality]... It's easier to do the virtual appointment, so I think... post-COVID, when you call in, they assume that you want a phone-related appointment. And if I'm talking to the desk people, I have to be like, 'I want to come in today.' [PWOUD19]

As a clinical tool, FP participants acknowledged that virtual visits were not suitable for all patients, with clinical complexity being a key determinant in the modality of care delivery:

> … a significant number of the patients that I serve have concurrent disorders in terms of psychiatric conditions and challenges as well, virtual care can be so challenging in that context...it feels like a very high barrier in that context. So I think when people are well, when the system's working well for people, when people are on stable doses of medications, when people are well managed in their substance use and primary care and mental health needs, then it makes a lot of sense to give the flexibility around having virtual support. [FP08]

Assessing the adequacy and appropriateness of virtual care itself, FPs emphasized that some conditions required in-person visits:

> [It] is a concern because we're not able to lay eyes on them. We're not able to do an exam. People with like cardiac conditions, respiratory conditions, it's harder to do those assessments and be on top of those conditions when they're just calling in. And people don't necessarily want to talk about other things. I think it's more of an opportunity in-person when you're with someone to be like, 'Hey, you know, it's really important that we address this.' Versus if they're over the phone, they're maybe not as receptive to wanting to manage some of those conditions. [FP07]

This suggests an interplay between system accommodation (i.e., patients' flexibility to choose a modality) and care adequacy (i.e., FPs' comfort with providing virtual care for particular conditions). While virtual care may be a more convenient modality, PWOUD participants also recognized its limitations and the value of in-person consultations:

> I'm busy. I have two kids and two jobs. So, I need to do those [prescription refill] appointments virtually. But of course, my other health conditions that I have … Like I have a heart problem. And then that gives me also a vein problem. I have problems with my legs because they're far from your heart. You know what I mean? So those are the conditions. Aside from that, I have a degenerative disc in my spine back. I'm like 53, and I've lived a hard life. So, I've got a few other like conditions that need in-person treatment. [PWOUD19]

Several participants noted that video-based virtual care could allow for a greater degree of personal connection and visual guidance for clinical assessments compared with phone-based visits. However, access to and provision of video-based appointments with PWOUD was limited by technological barriers, including access to video-enabled devices and usability challenges (experienced by both FPs and PWOUD): *"Often video conferencing, you're fiddling for five minutes, and then the appointment's over so it's just more efficient with our time and the patient's time to just do a phone call"* [FP13].

## Awareness

While all FP participants had experience providing virtual care to their PWOUD patients, not all PWOUD participants had direct experience with virtual primary care. In most cases, PWOUD awareness of virtual care was the result of having been offered a virtual appointment by their primary care provider so, for those who had not been offered this modality of care, awareness could be limited:

> I've never, ever heard of virtual care before, until I just came in here and read this, to be honest. But I just know that drug addicts alone, they're very…like they're mad at themselves for using. And they're uncomfortable going out in public…The virtual aspect of it could bring a lot of help to people that aren't getting help right now. You know, they don't want to go in to the doctor's. … You know, I know a lot of people that have phones and so I think this is actually a really smart way to go. [PWOUD01]

Conversely, for PWOUD participants who had experienced a virtual primary care appointment, some indicated that clearer communication was needed around when an appointment was going to be conducted virtually and when it was scheduled for in-person. This confusion could lead to missed appointments and frustration around a lack of respect for individuals' time.

## Discussion

We thematically analyzed interviews with FPs and PWOUD to explore their experiences with and perspectives on the impact of virtual care on access to primary care in BC, Canada. Guided by the six dimensions of access [49], our study provides insight into the interconnected and multifactorial influences on virtual primary care access for PWOUD. Overall, virtual visits were perceived by participants as a valuable option for accessing primary care, in addition to OUD management, but its use should be tailored to the needs and preferences of individual patients and providers to support primary care access.

In our interviews, systemic barriers such as health workforce shortages and limited provider capacity constrained primary care access, despite virtual care's ability to maximize practice efficiency [53,54]. Accessibility of virtual care services was a major benefit for PWOUD who experience transportation barriers and required a timely appointment. However, for PWOUD without consistent phone or internet access, inequities persisted. Participants appreciated the accommodating features of virtual care – specifically the convenience and flexibility of appointments – aligning with previous studies [26,55–57]. As noted by our participants, virtual care has the benefit of reducing disruptions to their lives, which is known to support OAT adherence [58,59]. Conversely, concerns remain about the use of virtual modalities and its clinical appropriateness for complex clinical indications, aligning with findings from a scoping review of strategies, benefits, and challenges of virtual primary care for PWOUD [26].

Participants noted that virtual care can reduce affordability barriers such as travel costs and wage loss for patients, despite additional set-up costs for providers, supporting previous findings [59]. Furthermore, virtual care is known to help connect with people in rural areas [26] or those experiencing homelessness [60–62]. However, for some PWOUD who experience multiple, intersecting socioeconomic inequities [63], inconsistent phone and/or internet access and lack of private spaces to hold confidential medical conversations can impede access to and acceptability of virtual care.

Virtual care was also seen as an option to reduce exposure to stigma and challenging environments, offering comfort for those accessing health services. However, many participants also valued in-person interactions for building trusting relationships with care providers and communicating nuanced health concerns [55]. These in-person visits are an important piece in relationship building and forming a therapeutic alliance [56,64], which greatly influence healthcare access. Both PWOUD and FP participants frequently felt that physical exams were crucial for positive health outcomes. Clinical appropriateness and virtual care's limits to performing physical exams often restricted the scope of care that could be provided virtually beyond that which FPs were confident dealing with by phone or through a video appointment [55,56].

Finally, awareness of virtual care varied. While some PWOUD participants were familiar with and actively engaged in virtual modalities, others were unaware that virtual primary care appointments were an option. This highlights the role that primary care providers themselves play in communicating care options to their patients, not to mention the importance of primary care attachment. Community-based outreach is well known to play a critical role in facilitating awareness and uptake of health services for PWOUD, underscoring the importance of tailored engagement strategies [65].

Our findings support a multifaceted approach to improve equitable access for virtual primary care for PWOUD, especially given the number of disparities that PWOUD face [65–70]. Interdisciplinary and collaborative primary care models should be encouraged to enhance the availability and clinical utility of virtual care for this patient population. Furthermore, health systems should invest in infrastructure and provider support to address persistent workforce shortages and technological gaps that limit patients' access to virtual care [29]. To address accessibility and affordability barriers, strategies such as subsidized phone and internet access and continued support for community organizations serving PWOUD can help mitigate disparities [71]. Importantly, virtual care should complement, and not replace, in-person care due to the value of a beneficial therapeutic relationship and comprehensive physical exams [55,56,72]. Thus, a hybrid model that supports flexibility of appointment modality based on patient preference and their primary care needs is recommended. Ultimately, tailoring virtual care to the complex and intersecting needs of PWOUD can support equitable access to primary care.

### Strengths and limitations

While this research adds important insights from PWOUD participants to the literature on virtual primary care, internalized stigma and the sensitive nature of substance use may have influenced participants' willingness to share openly, affecting how some experiences were reported. All interview participants are also subject to recall bias. While we recruited and conducted interviews with participants across BC, most participating FPs identified as women, most PWOUD participants identified as men, and the preponderance of participants resided in urban areas. As such, these findings may not be generalizable to other populations in BC and those in other jurisdictions with different regulatory environments that inform health service delivery and access. Nonetheless, the findings are particularly valuable given the challenges in capturing patient perspectives and are strengthened by our methodological rigour and reflexivity.

### Conclusion

This study explored the virtual care experiences of FPs and PWOUD to understand how it has impacted primary care access for PWOUD in BC. The findings highlight important dimensions of access that influence virtual care. As virtual care becomes a regular component of primary care, ensuring equitable access will require attention to all six dimensions of access – accessibility, availability, acceptability, affordability, adequacy, and awareness.

### Supporting information

**S1 File. Interview Guide – Family Physicians.**
(PDF)

**S2 File. Interview Guide – People with Opioid Use Disorder.**
(PDF)

**S3 File. Codebook.**
(PDF)

## Acknowledgments

We are grateful to the family physicians and people with opioid use disorder for sharing their time and experiences with our study team. The authors also extend thanks to Keaton Fraser for his thoughtful facilitation of our PWOUD interviews, as well as the broader VPC OUD study team for their support throughout the conduct of this research.

## Author contributions

**Conceptualization:** Lindsay Hedden, Rita Katherine McCracken.

**Formal analysis:** Shawna Narayan, Sarah Spencer, Sarah Muñoz-Violant.

**Funding acquisition:** Lindsay Hedden, Rita Katherine McCracken.

**Investigation:** Shawna Narayan, Sarah Spencer, Ellie Gooderham.

**Methodology:** Lindsay Hedden, Rita Katherine McCracken.

**Project administration:** Sarah Spencer.

**Supervision:** Lindsay Hedden, Rita Katherine McCracken.

**Writing – original draft:** Shawna Narayan, Sarah Spencer.

**Writing – review & editing:** Shawna Narayan, Sarah Spencer, Lindsay Hedden, Ellie Gooderham, Sarah Muñoz-Violant, Rita Katherine McCracken.

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
