## [Decision Letter · Decision Letter 0]

22 Jan 2026

Response to Reviewers
Revised Manuscript with Track Changes
Manuscript
**Journal Requirements:**
**Additional Editor Comments (if provided):**
**Reviewers' Comments:**

**Comments to the Author**

1. Does this manuscript meet PLOS Digital Health’s publication criteria?

Reviewer #1: Yes

Reviewer #2: Yes

Reviewer #3: Yes

2. Has the statistical analysis been performed appropriately and rigorously?

Reviewer #1: N/A

Reviewer #2: Yes

Reviewer #3: Yes

3. Have the authors made all data underlying the findings in their manuscript fully available (please refer to the Data Availability Statement at the start of the manuscript PDF file)?

Reviewer #1: No

Reviewer #2: Yes

Reviewer #3: Yes

4. Is the manuscript presented in an intelligible fashion and written in standard English?

Reviewer #1: Yes

Reviewer #2: Yes

Reviewer #3: Yes

Reviewer #1: The authors have provided a look at how persons with opioid use disorder might (or do) benefit from virtual care options. This is an interesting work and contributes to the knowledge building around virutal care in underserved populations. I suggest some minor revisions, and in particular emphasising the discussion around virtual care in the first section of the results section (accessibility). Please see specific comments below:

- Check spelling/grammar throughout paper - for example on line 84, 'barriers' is incorrectly capitalised, and British Columbia is used in both it's expanded and abbreviated state throughout

- In the methods section, it is mentioned that recruitment was pursued until sufficient data was collected (Lines 131-132). Perhaps you could expand on how you made this determination. You have supporting citations for the statement but I would like to know how the methods from these articles translated into how you actually determined you had sufficient data.

- In the data collection and analysis section, the lengths of the interviews (range and average) seem to be more results than methods. Consider moving this data to the results section

- It would be helpful to have your codebook as supplementary material if possible.

- For the first part of the accessibility section in the results, it is unclear how the first quote relates to virtual care. I'm assuming this doctor then sees those remote patients virtually, and this helps them keep these patients on the attachment list when they struggle to find care in their location, but this doesn't come through very clearly. Similarly with the last quote of the section (about the patient with MS), you briefly mention the advantage of virtual care for this patient, but it doesn't really come through clearly from the patient's quote or your writing. Perhaps be a bit more explicit with what the advantages are here. Finally, at the end of this section you say "the concept of a virtual appointment suggested to this participant that their health might improve" - this statement is not clear to me - is this something that you as the interviewer suggested to the patient or something that the patient came up with when learning about virtual care options?

- The quote from 278-284 and it's supporting information would benefit from a bit more discussion. Since the paper up to this point has mostly focused on positive aspects of virtual care, this aspect of requiring in-person visits in some cases despite the virtual offering could have more detail.

Reviewer #2: General comment: I thought this paper overall was in decent shape even in its current form. It would benefit from an edit pass and some minor revisions.

- Have other OUD studies used this same theory?

- Do other studies on virtual care delivery have similar findings to OUD?

Author Summary

- Pandemic is mentioned here although does not show up anywhere in the abstract

Introduction

Line 84 - Barriers capitalized

Sampling and recruitment

Line 125 - Is this typical to group residents with practising physicians?

Ethics and study rigour

Line 183-184 - Is the development of interview guides documented?

Line 183-184 - Did all interviewers hold a relevant degree in health sciences?

Discussion

- Can you back up the interview findings with any more journal references?

Reviewer #3: This paper seeks to understand the virtual care experiences of family physicians and people with opioid use disorders, relying on a thematic analysis of semi-structured interviews with both participant groups. Overall, the paper is well-written, methodologically rigorous and sound, and should be of interest to research community. I have a few small nitpicks that should be easy to address.

While the majority of the paper is well-written, the abstract is a bit terse in its writing style. The authors should consider some light edits to improve the flow here.

Were participants compensated for their participation in the study?

It would be nice to include some more details on which or at least how many authors conducted the interviews and analysis. Was it all authors, a subset, or perhaps even researchers not listed as authors included (I expect this isn’t the case, as I would expect this to be in the acknowledgements at least if it were the case)?

Line 84: Barriers is capitalized

While the Strengths and Limitations section does mention the fact that participants were recruited solely from British Columbia, it’s not really framed as a limitation. The fact that this might limit the applicability of findings to other geographic locations should be more clearly stated.

From skimming the references, I don’t see any references from the CHI/CSCW literature. I don’t work in this specific area (i.e., opioid use) so I’m not sure if there is work specifically on this, but there are certainly researchers doing qualitative work in similar spaces (i.e. healthcare access), so there might be. If not already done, it might be worth doing a quick search to see if there are relevant works to cite from those communities.

**Do you want your identity to be public for this peer review?** For information about this choice, including consent withdrawal, please see our Privacy Policy

Reviewer #1: No

Reviewer #2: No

Reviewer #3: No

**Figure resubmission:**

**Reproducibility:** To enhance the reproducibility of your results, we recommend that authors of applicable studies deposit laboratory protocols in protocols.io, where a protocol can be assigned its own identifier (DOI) such that it can be cited independently in the future. Additionally, PLOS ONE offers an option to publish peer-reviewed clinical study protocols. Read more information on sharing protocols at https://plos.org/protocols?utm_medium=editorial-email&utm_source=authorletters&utm_campaign=protocols

---

## [Editor Report · Decision Letter 1]

24 Feb 2026

“It could bring a lot of help to people that aren't getting help right now”: A qualitative analysis of the impact of virtual care on access to primary care for people with opioid use disorder

PDIG-D-25-00849R1

Dear Dr. McCracken,

We are pleased to inform you that your manuscript '“It could bring a lot of help to people that aren't getting help right now”: A qualitative analysis of the impact of virtual care on access to primary care for people with opioid use disorder' has been provisionally accepted for publication in PLOS Digital Health.

Best regards,

Girum Tareke Zewude, PhD

Academic Editor

PLOS Digital Health